# Impact of Microbiological Activity and Moisture on the Surface pH of Rock Art Sites: Cueva del Ratón, Baja California Sur, Mexico and Other Sites

**Ian Donald MacLeod [1,*] and Valerie Magar [2]**

1    Department of Materials Conservation, Western Australian Museum, Fremantle 6160, Australia
2    International Centre for the Study of the Preservation and Restoration of Cultural Property (ICCROM), 00153 Rome, Italy; valerie.magar@iccrom.org
\*    Correspondence: ian.macleod@museum.wa.gov.au; Tel.: +61-419952706

## Abstract

This paper reviews the apparent impact of how changes in nitrate, sulphate activities, and moisture affect the surface pH of rock art paintings at Cueva del Ratón, in the Sierra de San Francisco in Baja California Sur, Mexico. The data was collected after atypical weather events caused rain and mist in this normally arid area. The rock art paintings had been previously examined over several years and observed the unexpected formation of silica skins over some surfaces; such coatings are not often experienced in arid environments. The local geology of the cave and the availability of moisture can dramatically alter the microbiological activity on faecal material and development of surface acidity from such reactions which interacts with both the host rock and the pigments. Through a series of surface pH measurements and localised measurements on chloride, sulphate and nitrate it appears that both nitrate and sulphate concentrations have a significant impact on the surface pH, which is controlled by a diffusion-based movement of moisture from the closed to the open end of the shelter. The exfoliation of the rock surface and formation of the silica skins involves chemical reactions as contrasted with diffusion-controlled reactions which distribute the metabolites of the yeasts, moulds and bacteria, which are dominated by the availability of water through drip lines. The results are particularly relevant due to changing weather patterns in the last decade, caused by climate change, with an increase in hurricanes directly affecting the Baja California peninsula. The use of disposable test strips for semi-quantitative assessment of how these major anions impact on the decay mechanisms was a novel response to budget constraints and the remoteness of the location.

**Keywords:** rock art; Mexico; pH; moisture; microbiological activity; nitrate; sulphate; anion test strips

## 1. Introduction

    Conservation of rock art painted on walls of caves and overhangs in mountain chains has been the subject of intensive study for more than fifty years. Typically, the loss of surface detail will be due to a combination of exfoliation of the substrate to disbondment of the paint from the rock surfaces, due to changes in adhesion. In addition, there are issues associated with the cohesion of the paint layers themselves since biodeterioration can readily impact on the way in which the paint ties itself into a matrix that delineates the images. Painted surfaces inside overhangs are not subject to the change in microenvironment wrought by increased moisture and carbon dioxide levels that were destroying the famous Lascaux

caves [1]. They are however subject to changes in water activity wrought by alterations in the hydration profiles associated with changes in vegetation growing above and beyond the rocks holding the painted images. Sites on granite monoclines can also be subject to the impact of water being driven by cyclonic rains onto the painted surfaces which are normally free from direct interaction with flowing water [2] and from incoming fog during winter months. Many sites are in arid regions, and the paintings were made several thousands of years before the present time and it is the lack of moisture that has been a major factor in the preservation of the images, since water changes alters the cohesion and adhesion of the pigments on the rock surfaces. Apart from the changes in surface chemistry and the interactions of pigments with plant metabolites, increased water vapour results in overall increases in the levels of all microbiological activity. Bacterial respiration during their breeding and life cycles releases acidic metabolites which often interact negatively with the painted surfaces to alter the nature of the pigments which concomitantly changes the adhesion of the paint to the substrates and the cohesion of the pigments in the images.

The site at Cueva del Ratón had been the focus of an applied research programme for many years and several other researchers had studied the nature and form of the complex artwork which was painted within a time span ranging from 7000 to 1300 years ago [3,4]. A detailed description of the site was published in 2013 by Rubio which provides the most relevant details and summarises findings [5]. The region is characterised by a desert environment which had remained mainly unchanged since the Holocene, and receiving less than 100 mm of rain per year. However, the presence of silica skins over some of the images indicated that in previous times there had been periods with enough water present to activate bacteria which resulted in mobilisation of silica in the tuff formation and of the oxalate mineral whewellite [5,6]. During the week leading up to the site visitation to the Cueva del Ratón on 20 September 2015, it was uncommonly wet. As a result of various tropical storms in the Pacific Ocean, it rained substantially in this central part of the peninsula. As a result of these rain events, the landscape was totally transformed, turning green, with numerous cascades visible on cliffs, and with an increase in the flow of streams, which are otherwise usually dry or with extremely little amount of water in them. This increase in rainfall has been happening more frequently in the last decade, with an apparent northern shift in the hurricanes [6]. Owing to the long thin peninsula being sandwiched between two bodies of water, the impact of global warming on the local climate has been the subject of great scrutiny and recent review [7–9]. On the way from La Paz (at the southern end of the peninsula) to San Ignacio (in the central part), the road was destroyed by water and debris in one of these streams (Figure 1).

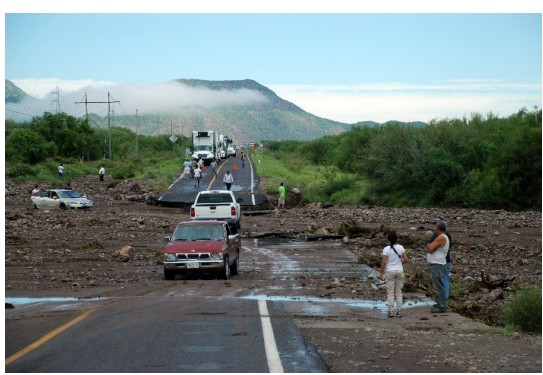 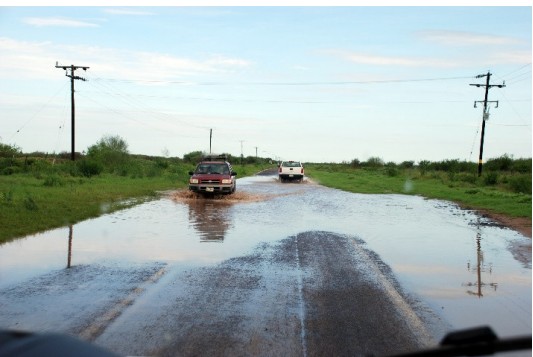

(**a**) Washed out road near La Paz        (**b**) Luxuriant growth and loose water

**Figure 1.** Impacts of unusual storms on the roads along the Baja California peninsula.

## 2. Description of Cueva Del Raton

Rock art found in more than 350 shelters in the Sierra de San Francisco illustrates the exquisite nature of the paintings and the cultural significance to the people of Mexico and to the wider community as an outward and visible manifestation of the inner sense of place and identity that resulted in the intellectual and spiritual outpouring of remarkable creativity that depicted the local flora and fauna [6,10]. The mainly realistic large-sized images led Crosby [11] and Hambleton [12] to coin the term Great Murals in the 1970s. The area was inscribed on the UNESCO's World Heritage list in 1993. Cueva del Ratón, located on a shelter at 1100 m above sea level, in the higher part of the mountain range—unlike most other sites which are located at the bottom of deep canyons, contains human and animal representations as well as some geometric motifs, painted in red, black, white and yellow, with often dense superpositions of figures, located in different areas of the shelter. The painted rock art images at Cueva del Ratón and the nature of the mountain landscape are shown in Figure 2. The site has been protected by a fence since the 1990s, to avoid goats raised by nearby villagers from entering. However, concentrations of goat guano are still visible in some areas of the shelter and the impact of this material on the biological decay mechanisms is described.

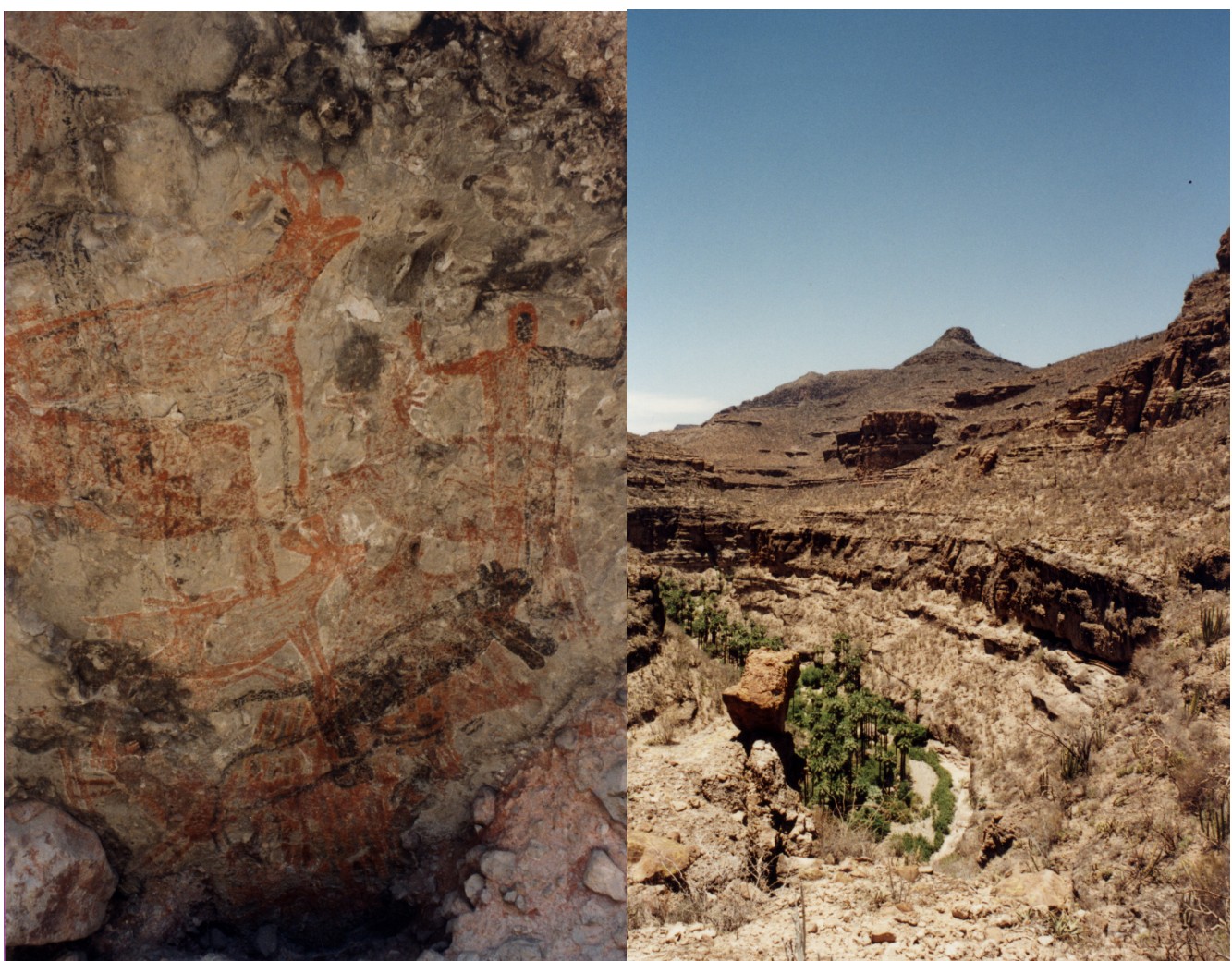

(**a**) Detail of paintings at Cueva del Ratón          (**b**) View of the high Sierra landscape

**Figure 2.** View of the rock art and landscape of the Sierra de San Francisco.

Today on rock art sites around the world the value of the images only increases as time goes by since they reflect an environment of which we know little. It is thus vital to understand the physical processes of deterioration so that appropriate conservation management of the areas can be based on good understanding of not only the material science but also of the intangible heritage values. In times of major climate change the rock art images recorded in Spanish and French caves, across Australia, the whole North American continent and in South America provide a vital link to what the changing climate was doing to the supply of food and to the way in which the people were responding to their environment. Rock art is as far removed from being the esoteric concern of a small group of academics, well known in concentric circles, as the reader is from the mechanical and electronic processes that brings research articles to their screens.

Conservators use the precise identity of original and altered pigments as their anatomical charts to determine what the nature of the sites was like when the images were created and how the changing environment has impacted on the art works. The precise kinetics of how pigments have altered can be a matter of conjecture but indirect dating methods through studies of the accretions covering parts of the paintings can provide a useful guide as to the dates when events took place [13]. However, there are many technical difficulties in using $C^{14}$ dating of pigments and accretions which are readily "contaminated" by recently deposited oxalate deposits.

## 3. Materials and Analytical Methods

Except for charcoal as a pigment, the colourants used in rock art paintings are minerals which have been ground into a fine powder and mixed with some form of binder. The binder can be as simple as saliva or blood or composed of juices extracted from roots of plants [13–15]. Perhaps the best example of a detailed analysis of how pigment sizes control the rates of deterioration in a very complex pattern of competing forces of chemical and biological attack on the bonds between pigment and substrate was published in 2023 by Ozan et al. [16] Apart from the biological attack on the binders the reactions of the pigments are controlled by a series of acid-base reactions, which tend to mobilise the metal ions or change the anions in the minerals into a more soluble form. Examples of the latter are adsorption of acidic moisture by carbonates to turn the minerals into soluble bicarbonates which then run away from the painted area by gravity or some diffusion-controlled process. Many of the pigments are metal oxides or hydroxy-oxides and under suitable conditions they react with the acidic solutions and lose the original bonds with the substrates [17]. The acidity of a solution or a rock surface is assessed by placing a flat surface pH electrode against the surface and instilling a drop of water between the electrode and the surface. It normally takes $60 \pm 5$ s to obtain a steady surface pH but if the surface was more responsive and the pH reading stabilised in 40 s, then that value was recorded. Prolonged equilibration times can result in pH values that are not reflective of the local microenvironment since the instilled water droplets can be drawn by capillary action into the rock or pigment surfaces. Portable flat surface pH electrodes consist of an internal Ag/AgCl reference electrode which connects with the surface under test with fine wicks that provide electrical connectivity. The self-ionisation of water has an equilibrium value of $10^{-14}$ so the pH value, negative logarithm of the hydrogen ion concentration for strong alkali solutions is 14 and for strong acid it is zero.

When metal hydroxides are mobilised by acid dissolution the generic dissolution reaction can be written in the form shown in Equation (1),

$$M(OH)_n + n\,H^+ \rightarrow M^{n+} + n\,H_2O. \tag{1}$$

In Equation (1) the *n* value is the oxidation state of the metal, typically 2 and 3 for iron and mixtures of 2, 3 or 4 for manganese. Similar equations can be written for oxyhydroxides such as all the iron minerals of the form FeO.OH, which covers goethite, hematite, maghemite and lepidocrocite. The pH electrode was calibrated with standard pH buffers at pH 4 and pH 7 before on-site measurements commenced. Tests on the pH electrode showed that the calibrated values were stable for more than a week [18]. The pH The spelling should be meter which defines the type of instrument and not a unit of measurement of distance was temperature compensated using a thermocouple connected to a Crison pH metre and the pH electrode was a VWR model no W7567287. For distilled water equilibrated in air the pH values are typically $6.6 \pm 0.2$, due to carbon dioxide from the atmosphere creating a very weak bicarbonate solution. Iron minerals have natural pH values of $5.4 \pm 0.2$ due to hydrolysis reactions such as shown in Equation (2), which is closely followed by reaction number three.

$$Fe_2O_3 + 3\,H_2O \rightarrow 2\,Fe(OH)_2^+ + 2\,OH^- \tag{2}$$

$$Fe(OH)_2^+ + 3\,H_2O \rightarrow Fe(OH)_3 + 2\,H^+ \tag{3}$$

The surface pH values across the Cueva del Ratón site are listed in Table 1 and not surprisingly the mean pH of the iron pigments was $5.12 \pm 0.33$ coincides with the natural rock pH of iron(III) minerals. Area B3-6 had a surface pH of 6.89 which is significantly higher (more alkaline) than most of the other values and is associated with the freshly exfoliated surface, which is rich in bicarbonate minerals (Figure 3).

**Table 1.** Location of pH and anion measurements (ppm) at the cave site, September 2015.

| Location | Description | pH | $SO_4^{2-}$ ppm | $Cl^-$ ppm | $NO_3^-$ ppm |
|---|---|---|---|---|---|
| B1-1 | Red pigment | 4.77 | | | |
| B1-2 | White surface on unit 1 | 4.87 | | | |
| B1-3 | Black material (on red pigment) | 5.11 | | | |
| B3-1 | Loss area | 5.38 | | | |
| B3-2 | Loss area (same loss area as B3-1) | 5.59 | 300 | 0 | 10 |
| B3-3 | Red small animal | 5.65 | | | |
| B3-4 | Yellow pigment | 5.57 | | | |
| B3-5 | Unit 1, on white material | 5.46 | | | |
| B3-6 | White salt on loss (see area B3-1) | 6.89 | 400 | 1500 | 25 |
| D1-1 | Circular loss area, centre | 4.41 | 100 | 1000 | 175 |
| D1-2 | Circular loss area, edge | 4.62 | | | |
| D1-3 | Red with black covering | 5.02 | | | |
| D1-4 | Exfoliation | 4.59 | trace | 2000 | 500 |
| D1-5 | Below round | 5.31 | | | |
| D1-6 | Exfoliation | 5.02 | | 2500 | 175 |
| D1-7 | Yellowish | 5.09 | | | |
| D1-8 | Black | 5.24 | | | |
| E1-1 | Stable part | 4.48 | | | |
| E1-2 | Loss with white | 4.92 | | | |
| E1-3 | Loss with red | 4.87 | | | |
| E1-4 | Unit 1 | 4.83 | | | |
| F1 | Silcrete skin | 4.41 | | | |
| F2 | Loss of skin | 4.48 | | | |
| F3 | Interphase between loss and skin | 5.01 | | | |

Although the pH is an excellent guide to the inherent reactivity of surfaces, it is often helpful to know what other variables are controlling the biological activity and mineral reactivity on the rock surfaces. Experience in monitoring rock surfaces in the similarly hot and arid Pilbara region of Western Australia had shown that when the pH is significantly less than the average value of iron minerals it is associated with soluble nitrates being used by microflora as an energy source [18,19]. In the Pilbara rock art studies, the amount of chloride, nitrate and sulphate on rocks is assessed by washing rocks with ultrapure water

and using ion chromatography to quantify the values. Given that there were no facilities that could analyse the surface washings and that the images were on vertical or overhead horizontal areas of the cave, alternative assessment methods for gauging the impact of the common chloride, nitrate and sulphate anions were used.

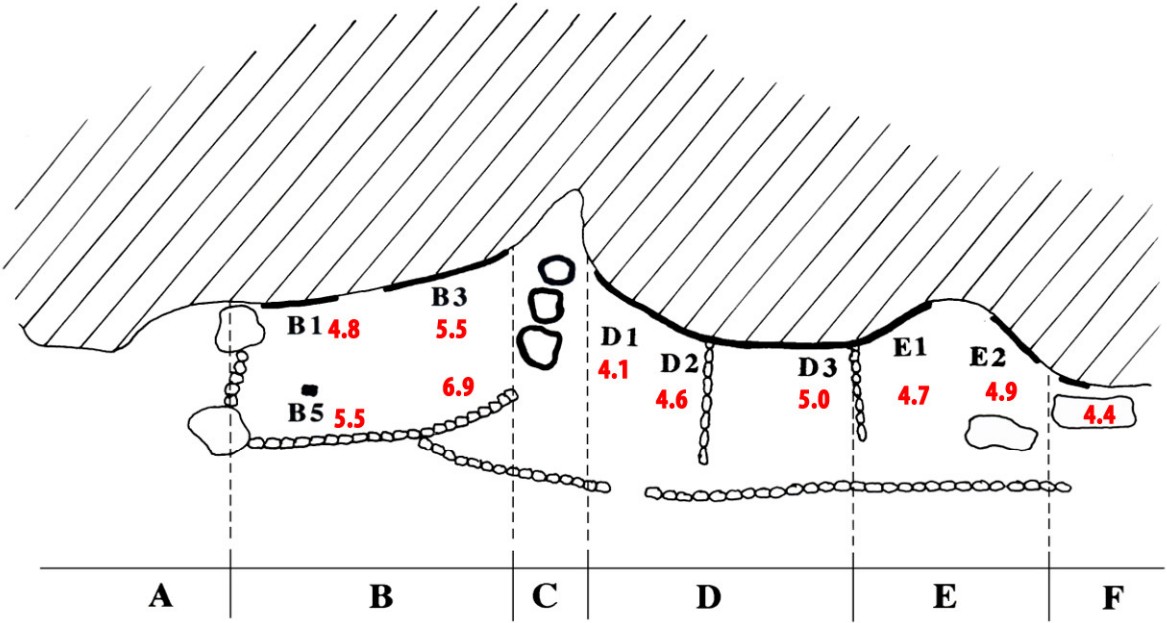

**Figure 3.** Plan view of the Cueva del Ratón site with typical pH values.

The Merckoquant® test strips are colourimetric indicators of ion concentrations. The ranges are $1600 < SO_4{}^{2-} > 200$ mg/litre and $500 < NO_3{}^- > 10$ mg/litre and for chloride ions the range is $3000 < Cl^- > 500$ mg/litre. By moistening the test strips with distilled water and holding the sensing sections against the surfaces for a minute, the strips indicated by the intensity and chroma of the colours what the levels of anions were. It was possible to interpolate the intermediate values of the colours when the readings lay in between the concentration bands marked on the test strips. The advantage of the test strips was that there was minimal impact on the already friable painted surfaces and that it provided the authors with a guide as to the variable nature of the painted surfaces across the site. The data elucidated possible decay mechanisms that were controlling the site at Cueva del Ratón.

The impact of the heavy rain was also seen in the shelter of Cueva del Ratón where the changes in moisture content resulted in significant increases in the amount of plant growth [6–8]. The shelter is mainly constituted by two geological layers, both composed by tuff: unit 1, located in the lower part of the shelter (which has eroded, allowing the creation of the shelter), and unit 2, on which most of the rock art is located, usually on top of a case-hardened surface (Figure 4).

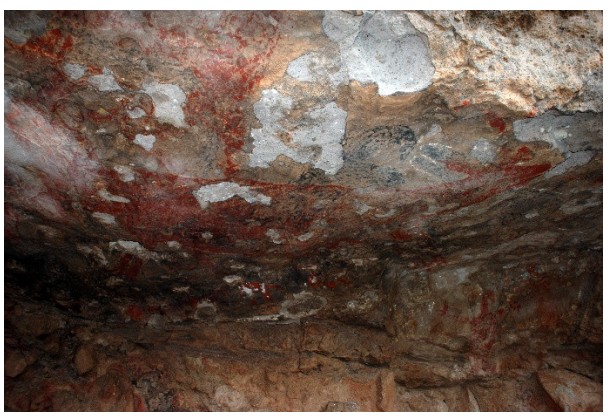

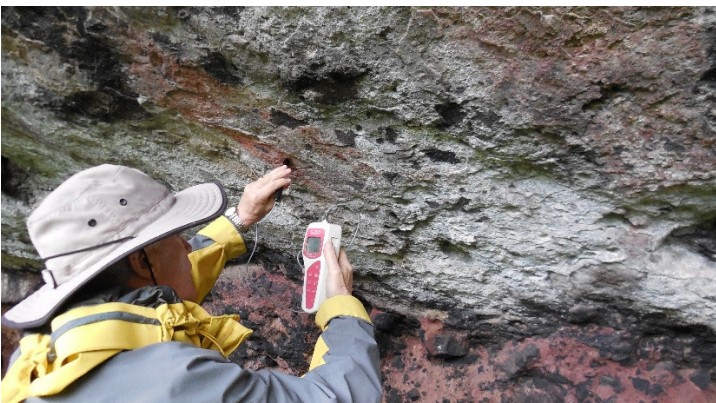

(**a**) Deer image and exfoliation in shelter   (**b**) Measurement of surface pH on site

**Figure 4.** Detailed views of Cueva del Ratón.

A series of pH measurements were made in four areas with rock art figures (Figures 2a and 3) with additional data collected on adjacent zones that showed various forms of accelerated decay, such as disbondment of the substrate. Firstly, the B1 section was the deepest and the highest part of the shelter, which was characterised by abundant black material on the surface, obscuring the rock art. The black material is likely to be due to formation of the black mineral whewellite, a hydrated form of calcium oxalate viz. $CaC_2O_4.H_2O$, where the oxalate comes from plant metabolites [10,11,15,20]. To obtain data from the main rock art panel, designated as B3, involved climbing up rocks to reach the surfaces for measurements (Figure 4). Standing on a small ledge, the D1 area was characterised by circular shaped losses. The two areas in the bottom part of the site, one with a slight protection (E1) and a vertical part (F). During the pH survey assessment of the sulphate, nitrate and chloride ions were also made using Merck Quantab test strips, as described in the section on methodology. Previous work in the Murujuga (Burrup) engraved sites in the Pilbara region of Western Australia had shown that there was a strong correlation between the nitrate concentration and more acidic surface pH values [18]. In these cases, the rocks were almost horizontal which facilitated irrigation techniques, and the eluates were analysed chromatographically which provided quantitative data.

## 4. Results

The results with a summary of the pH and the anion measurements are found in Table 1, with comparative results from measurements made on granite and limestone rocks located in Western Australia shown in Table 2. It is noted that while the climate and substrates are different, the Perth based data is useful to illustrate the phenomena in Baja California in that it shows the relationship between increased nitrate, as measured with the Merckoquant strips, and the pH measured with the flat surface electrode. Although the data on the concentration of the surface anions is limited at Cueva del Ratón, it was possible to discern direct connections between the amount of nitrate and the surface pH, as shown in Figure 5. For comparative purposes, the surface nitrates on the soil, which had a fulsome deposit of goat guano, gave a high nitrate reading of 500 ppm and the shiny patina recorded a 250 ppm value on the test strips being used to detect nitrate ions.

**Table 2.** Descriptive statistics on the distribution of pH values at Cueva del Ratón and Perth, Australia.

| Statistical Variable | Cueva del Ratón | Perth Limestone |
| --- | --- | --- |
| Mean | 5.07 | 4.80 |
| Standard Error | 0.11 | 0.08 |

**Table 2.** *Cont.*

| Statistical Variable | Cueva del Ratón | Perth Limestone |
|---|---|---|
| Median | 5.02 | 4.81 |
| Standard Deviation | 0.53 | 0.24 |
| Sample Variance | 0.28 | 0.06 |
| Kurtosis | 4.55 | −1.22 |
| Skewness | 1.59 | 0.06 |
| Range | 2.48 | 0.70 |
| Minimum | 4.41 | 4.48 |
| Maximum | 6.89 | 5.18 |
| Count | 25 | 9 |

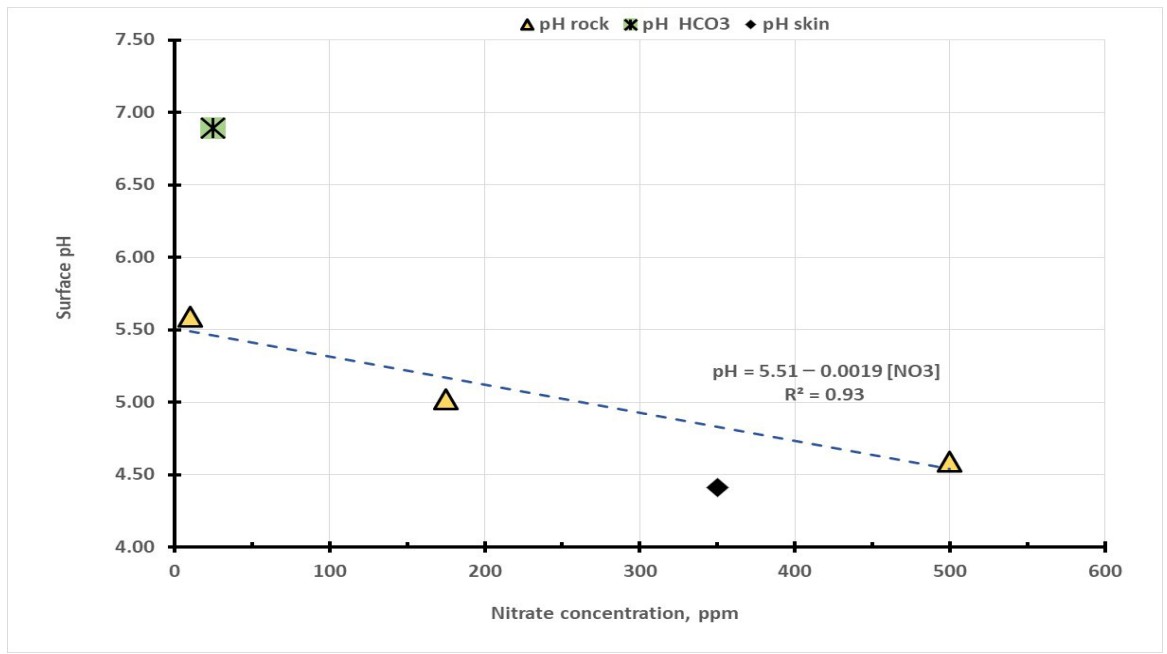

**Figure 5.** Plot of surface pH and nitrate ions, including exfoliation at Cueva del Ratón.

The pH data in Table 1 is plotted in Figure 5 as a function of the nitrate concentration from the Quantab test strips. At zero nitrate concentration (the intercept value in Equation (4)) the surface pH was 5.51, and the pH fell by 0.19 per hundred-fold increase in the nitrate concentration. The change in pH represents an increase of 155% in surface acidity per 100 ppm increase in the nitrate concentration (Table 1).

$$\text{pH}_{\text{Cueva del Ratón}} = 5.51 - 0.0019 \, [\text{NO}_3^-] \tag{4}$$

This intercept value is typical of the natural level of acidity associated with hydrolysis reactions of iron minerals with water (see pH values in Figure 3 at B3 and B5). The low nitrate levels found near the mouth of the cave are shown in the left side of figure The most alkaline pH on the site was at site B3-6 at 6.69 which is where there were white crystals associated with the surface loss. This pH is typical of recrystallized $NaHCO_3$ solution which has a $pK_{a1}$ of 6.4 for carbonic acid. Similar surface pH values were found on sites in the calcareous Napier Ranges in the west Kimberley region of Western Australia and the crystals were identified by X-ray diffraction as $NaHCO_3$ [10,11,13].

In reviewing the pH and anion concentration data collected in the Cueva del Ratón there was no comparable data from Australian rock art sites, since this was the first use of the methodology involving anion test strips on rock art sites in Mexico. The lack of

comparative data was overcome through a series of calibration tests using the same brand of sulphate, nitrate and chloride test strips on both granite and limestone rocks located in suburban Perth, Western Australia. This comparative study was conducted in September 2024 (spring season) as part of the review of the Mexico data collected ten years previously. Tests on a weathered granite rock used the same model surface pH electrode and metre used in Mexico. Since the pH calibration data in Perth gave the same responses with the flat surface electrode as on the Cueva del Ratón site, this was considered a good indication of the reproducibility of the pH measurements. This Perth based data showed that the surface pH fell with increased nitrate concentrations, in a similar way to the measurements in Mexico. Laboratory tested nitrate levels recorded from surface irrigation studies on Pilbara rock engraving sites in Western Australia also showed the pH fell with increased nitrate concentrations. In an upper area of the Perth calibration granite rock, which was better drained and more exposed to the sun, the pH was 5.56 with a nitrate concentration of between 10 and 15 ppm. This pH is typical of the acidity of the iron and manganese minerals that exist on the surface, where the minerals partially dissolve in the distilled water and produce acid solutions owing to hydrolysis of the metal ions, as described in Equations (1)–(3). Lower down the rock where moisture pooled in a crack, the pH was 4.94 and the nitrate level was correspondingly higher at 30–40 ppm. This data supported the measurements made at Cueva del Ratón, which are shown in Equation (4). The principal source of nitrate in Perth comes from air pollution associated with nearby automobile traffic as the chosen location was adjacent to two suburban bus routes.

The veracity of the strip measurements was also evaluated on a calcareous rock in the rear garden of the same home in suburban Perth. These tests showed that the chloride and sulphate ion concentrations did not affect the pH, but the Perth rocks were more acidic at higher nitrate values. The mean pH of the rock at Cueva del Ratón was $4.79 \pm 0.23$ which is indicative of microbiological activity (Figure 5). A comparison of the statistical distribution of pH on the volcanic tuff of Cueva del Ratón and the limestone rock is shown in Table 2. The data on the distribution of the pH values on the Perth limestone rock shows a very low range of values and this is because of its small size, approximately 20 cm $\times$ 20 cm, and 45 cm high, the rock was a moderately fairly uniform chemical environment.

The Cueva del Ratón site had a very peaked distribution around the mean value with a kurtosis value of 4.55 whereas the Perth rock was flattened in the pH distribution with a kurtosis of $-1.22$ (Table 2). The skewness of the Mexican site was significantly to the right side of the mean whereas the Perth rock was symmetrically disposed around the mean. The two sites had remarkably similar minimum pH values and as previously noted, the maximum pH reading at the cave was at a freshly exfoliated site that was consistent with freshly crystallised sodium bicarbonate. A plot of the surface pH on the suburban limestone rock and the nitrate concentration recorded on the test strips was characterised by the linear regression equation for the relationship between nitrate and pH is given by Equation (5).

$$\text{pH}_{\text{limestone rock}} = 4.97 - 0.01\,[\text{NO}_3{}^-] \tag{5}$$

It was noted that the pH of the limestone rock was five times more sensitive to nitrate than the volcanic tuff of the Cueva del Ratón site.

When the five sulphate ion concentrations shown in Table 1 are plotted against the pH it is evident that the acidity falls with increasing sulphate, which indicates that the alkaline solutions containing the sulphate ions are likely to be associated with the formation of the tuff rock and not have come pollutants. The linear regression (Equation (6)) is shown below,

and the $R^2$ for Equation (6) was 0.968 with a mean slope of 0.0054 ± 0.0006 pH/ppm $SO_4^{2-}$ (an error of ±11%) and an intercept pH of 4.37 ± 0.13, which amounts to a 3% error.

$$\text{Cueva del Raton } pH_{surface} = 4.37 + 0.0054\,[SO_4^{2-}]_{ppm} \tag{6}$$

A possible explanation for the greater sensitivity of the pH of the suburban limestone to nitrate ions is due to the absence of sulphate in the limestone rock, whereas at the Cueva del Ratón has different geology which produces alkaline sulphate ions that tend to mask the acidity associated with the presence of the nitrate ions. This would explain the five-fold greater sensitivity of the limestone reference rock to nitrate, compared with the rock art site in Mexico. The alkaline nature of the moisture that mobilised the sulphate from the parent rocks indicates that this process acts as a buffering mechanism for the increasing acidity arising from the microbiological utilisation of the increased nitrogen sources emanating from the goat guano. The pH response to sulphate ion concentration represents a decrease in acidity by a factor of 3.5 times for each 100-fold increase in sulphate ion activity. The dehydration and rehydration cycles of sodium sulphate can have a pronounced effect on rock weathering which may be one of the underlying causes of surface exfoliation at the Cueva del Ratón site [21]. The acidity of the intercept value in equation six reflects the impact of microbiological activity at zero sulphate levels.

The outlying pH measurement on the limestone test rock came from a grey-black section of patinated rock that was consistent with the formation of whewellite, i.e., $CaC_2O_4 \cdot H_2O$ which is very abundant on the calcareous sites in the Napier Ranges in the Kimberley region of Western Australia [13–17]. Although the extensive black deposits at the rear of section B3 of the cave at Cueva del Ratón site was not sampled during the site visit, it is likely that the black deposits noted in Table 1 at sites B1-3 and D1-3 are due to reactions that have formed deposits of whewellite. Samples analysed at the end of the 1990s had shown the presence of whewellite in both areas, and in B1, also elevated levels of manganese in the black deposit, and presence of sulphur. The similarity of the pH response to the nitrate concentration on both sites gives confidence that the data collected in Mexico has relevance to developing an understanding of the biodynamics of the site in California de Sur. A visual representation of how the sulphate and nitrate levels change the apparent stability of the rock surfaces, and the paintings is seen in Figure 6 which shows details of the black deposits in area B and the significant exfoliation problems that were reported on the walls in Section D (see Table 1 and Figure 4).

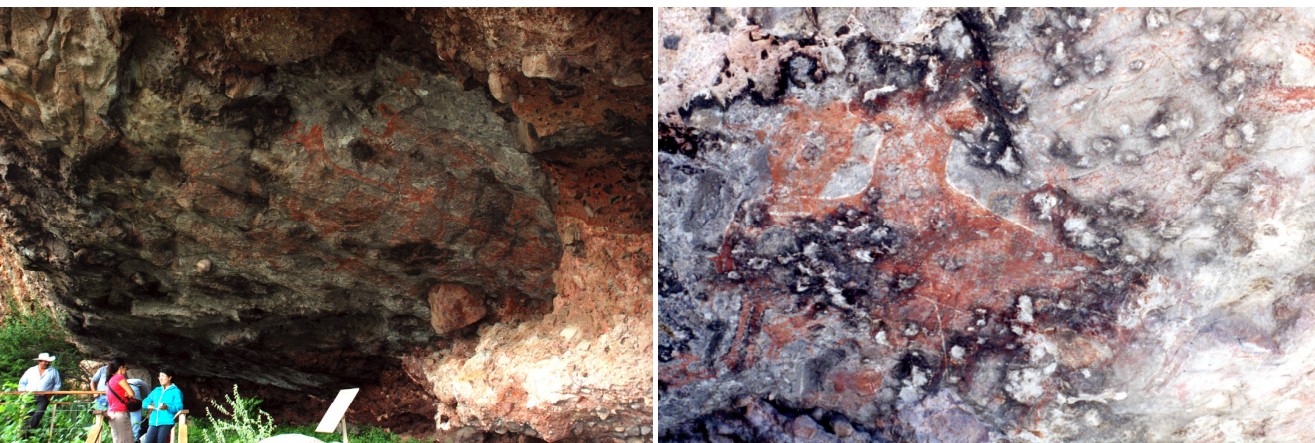

**Figure 6.** Left image shows area B with low nitrate at mouth of cave and the right image shows area D with high nitrate and exfoliation.

Anion concentration data were obtained from washings on Murujuga rock art sites were based on ion chromatography analysis using ultra-pure water. The concentration of nitrate was checked against the local surface pH measurements which demonstrated that there is a direct link between the minimum pH (most acidic) and the $NO_3^-$ found on the rock surfaces [17,18]. The bioavailability of nitrogen limits microbiological activity and since metabolites are acidic, it is not surprising to find a logarithmic relationship between increased microbiological activity and nitrate concentration. One unexpected correlation was that the concentration of chloride ions, from wind borne sea salts [21,22], has a major impact on adsorption of $NO_x$ released into the atmosphere by nearby industrial activities associated with processing of offshore natural gas for export.

## 5. Acidity Changes with Site Location

The spatial distribution of the pH values across the Cueva del Ratón site with the previously collected data on the depth, width and height of the cave opened the possibility of being able to establish if there were any systematic changes with the acidity values with location that might be off assistance in development of an appropriate conservation management plan for the site. With a collection of twenty-five sets of pH measurements across the site, from Area B to Area F (Figure 4), the pH data was plotted as a function of the distance variables across the site, as it was traversed from east to west. The data shows common linear trends of increasing acidity as the shelter becomes deeper and higher, and the point measurements are linked with each other in each transect, as seen in Figure 7. Higher acidity is consistent with less air movement in the inner parts of the cave, compared with much higher exchange rates near the opening of the cave.

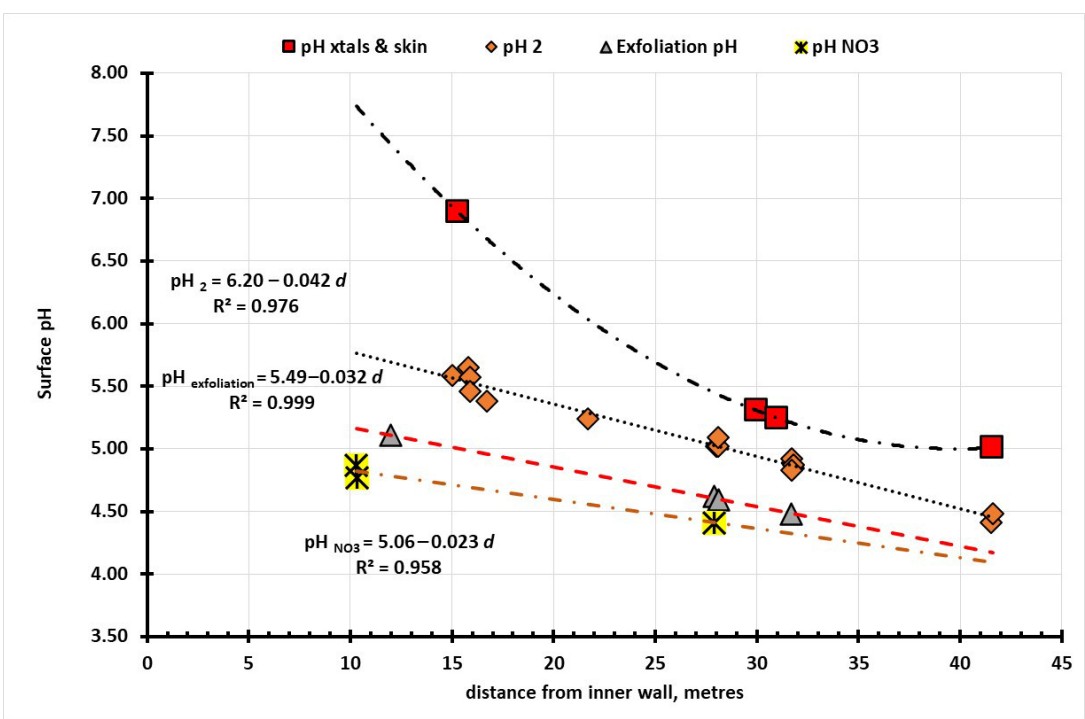

**Figure 7.** Variation in surface pH with distance from the eastern edge of the site.

For nitrate rich zones, the first three measurement points show the pH decreasing by 0.023 pH per metre or about 5.4% per metre. For the sections of the site with the greatest exfoliation the pH decreases by 0.032 per metre or around 7.6% and the upper most (highest) group the pH fell at the rate of 0.042 per metre or about 10% per metre. The results are summarised in Table 3. Analysis of the dependence of the slopes on the

distance across the site shows that each of the subsets are statistically significantly different to each other, which supports the belief that the microenvironment progressively changes in moving across the site. The linear changes in the pH with distance from the rear of the cave, for all but surfaces with crystals and skins, shows that the same mechanism is involved, albeit each area had different pH sensitivity to distance. This variable can be understood by the observation that as the Cueva opens to a more open structure the exchange rates of gaseous materials with the external environment will change in accord with increased air movement. It was also noted that the exfoliation sub-group is less acidic than the locations associated with higher nitrate concentration. This analysis shows that there are two different mechanisms controlling the pH in the Cueva del Ratón site. One is diffusion controlled, with linear changes in pH for alterations in distance and the other mechanism, which shows logarithmic dependence on distance, is chemically controlled.

**Table 3.** Sensitivity of pH to distance in traversing the site from east to west.

| Intercept pH | Slope (pH/m) | Slope (pH/log m) | $R^2$ | locations | Comments |
|---|---|---|---|---|---|
| 5.06 | $-0.023 \pm 0.005$ | | 0.958 | B1-1. B 1-2 D1-1 | Nitrate |
| 5.49 | $-0.032 \pm 0.001$ | | 0.999 | B1-3, D1-2, D1-4, E1-1 | Exfoliation |
| 6.20 | $-0.042 \pm 0.002$ | | 0.976 | See list below * | General |
| 7.64 | | $-4.59$ | 0.957 | B3-6, D1-5, D1-7, and F3 | Crystals and skin |

* The pH measurements made in areas B3-1 to B3-5, sections D1-3 and D1-6 to D1-8, E1-2 to E1-4 and in both F1 and F2.

The disbondment mechanism is consistent with the crystallisation of $NaHCO_3$ from solution weathering processes in the tuff on which the artwork has been painted. Although sodium bicarbonate is moderately soluble it will readily crystallise under drying wind conditions that often prevail in rock art shelters. Reviews of the mechanisms of deterioration of stone substrates by Doehne [23], Price [24] and Young [25] support the proposed deterioration mechanisms outlined above for the Cueva del Ratón site and should be read before attempting to model the deterioration processes that might be found on other rock art sites. Although the cultural practices and local climates will be different the fundamental nature of the geochemistry of the parent rocks is the foundation on which such interpretations should be based.

## 6. Discussion and Insights from Other Studies

Since experience across more than forty sites within the Murujuga National Park in the Pilbara region of Western Australia had shown significant relationships between acidity and nitrate levels, the data listed in Table 1 was checked for correlations between the surface pH and the chloride and nitrate test strips. These sites had shown the linkage [18] between elevated nitrate and pH values that were significantly lower than the acidity associated with simple hydrolysis of the iron-rich weathering crust. Data collected in the Murujuga park in the period 2003–2004 had shown a direct connection between the nitrate and the chloride ion concentrations, which implied that the wind-borne sea salt deposits were directly involved in the adsorption mechanism of $NO_x$ from the atmosphere. It was not surprising that for the Cueva del Ratón site there was a similar response of the nitrate to the chloride ion concentration, as shown in Equation (7),

$$^{\text{Cueva del Raton}} [NO_3^-]_{ppm} = 0.25 \, [Cl^-] - 17. \tag{7}$$

The $R^2$ value for Equation (7) was very high at 0.966, which is why the error in the slope was only $\pm 0.05$ for the locations (Figure 5) of B3-2, D1-1, and D1-4, but the intercept error was $\pm 60$ ppm. This means that when there is no measurable chloride concentration

the nitrate concentration will fall to zero, which indicates that the saltiness of the surface assists the rock in adsorbing nitrogen oxides from the local environment. Bischoff, Hiar, and Turco evaluated the use of nitrate test strips and showed that with experienced operators that the results compare well with other laboratory methods [26]. There was insufficient nitrate and chloride readings on the reference limestone rock in Perth to be able to discern a similar relationship. However, when 48 wash solutions from the Murujuga sites in February 2004 showed that the nitrate concentration was $0.11 \pm 0.01$ times the concentration of the chloride. The coefficient of variation in the wash solution ratios was 11% and the limited data from Cueva del Ratón test strips showed a variation in more than 350% which is consistent with a much smaller number of samples and the inaccuracy of interpreting the intermediate readings of when the colour changes in the test strips were in between specific levels.

The limited data at Cueva del Ratón indicates that the accumulation of salt deposits on the cave wall acts as a concentration factor for the adsorption of nitrates from the atmosphere or from nitrogen generating microorganisms. The large mass of goat fasces undergoing biodegradation at the front of the cave is likely to be a principle local source of nitrates. Surface washings on a remote island in the Dampier Archipelago in Western Australia showed significant amounts of nitrate that were collected within a metre of animal scats. The presence of a colony of bats in a cave in the Napier Ranges in Western Australia showed that they mean acidity was twice that of a nearby shelter that had no bats.

In the absence of ready access to a sophisticated chemical analysis facility, and the necessary funds to cover the costs of the work, it was thought that it would be useful to establish the relationship between surface chloride measured with a Cl-ion specific electrode and the Quantab test strips. Moistened test strips were placed on the limestone reference rock in Perth, and the results were recorded before the surface chloride ion activities at the same locations were measured with an Orion Cl-ion combination electrode, attached to a TPS multi-function digital metre. The electrode had been calibrated with 1000 and 100 ppm standards before measurements on the site. The data confirmed that there was a simple relationship between surface readings taken with a chloride ion specific electrode and the amount of salt assessed using the Merk test strips. The linear regression analysis results are summarised in Equation (8), which had an $R^2$ of 0.92 with an error of only $\pm 0.030$ in the slope which represents a 20% coefficient of variation in the sensitivity of the readings.

$$^{\text{Orion}} [Cl^-]_{\text{surface ppm}} = 0.145 \, [Cl^-]_{\text{Quantab}} - 4.01 \tag{8}$$

There is a large error of $\pm 7.0$ or 174% in the intercept, which means that it is unwise to use the equation to calculate the surface chloride concentration from the Quantab value. When the calibration factor of 0.145 (the slope in Equation (8)) is used to correct the Quantab values from Cueva del Ratón, the uncalibrated mean value of $1750 \pm 645$ ppm (Table 1) falls to $254 \pm 94$ ppm. The most likely source of chloride on the site is from the accumulation of salt spray carried inland by the prevailing sea breezes, for as the height of the ranges increase to around 1100 m, the fog and mist will allow condensation of sea salts. The site is less than 40 km from the ocean, as the crow flies [10]. Even at this salt level there is enough hygroscopicity to facilitate adsorption of nitrogen compounds emanating from abundant goat guano.

Thanks to the site mapping conducted by Parks Canada (Figure 8) for the joint project by the Getty Conservation Institute, the National Institute of Anthropology and History (INAH) and Amisud [27] in the early 1990s, the whole area was broken into eight sections which correspond to the numbered plan view of the site from B to F, as no surface measurements were taken on part A or part C of the site (see Figure 4). The primary difference between the way in which the pH responded to the increasing distance from the opening

of the shelter is that the area associated with the crystallisation of materials on the surface, leading to loss of the parent rock patina, or to the formation of silica skins, is that these reactions show a logarithmic dependence on distance which is consistent with a chemically controlled reaction. The different mechanisms are reflected in Figure 7 which illustrates that the three linear plots of pH increasing linearly with distance are distinctly different to the logarithmic dependence of the areas in which there are crystalline deposits and areas associated with silica skin formation.

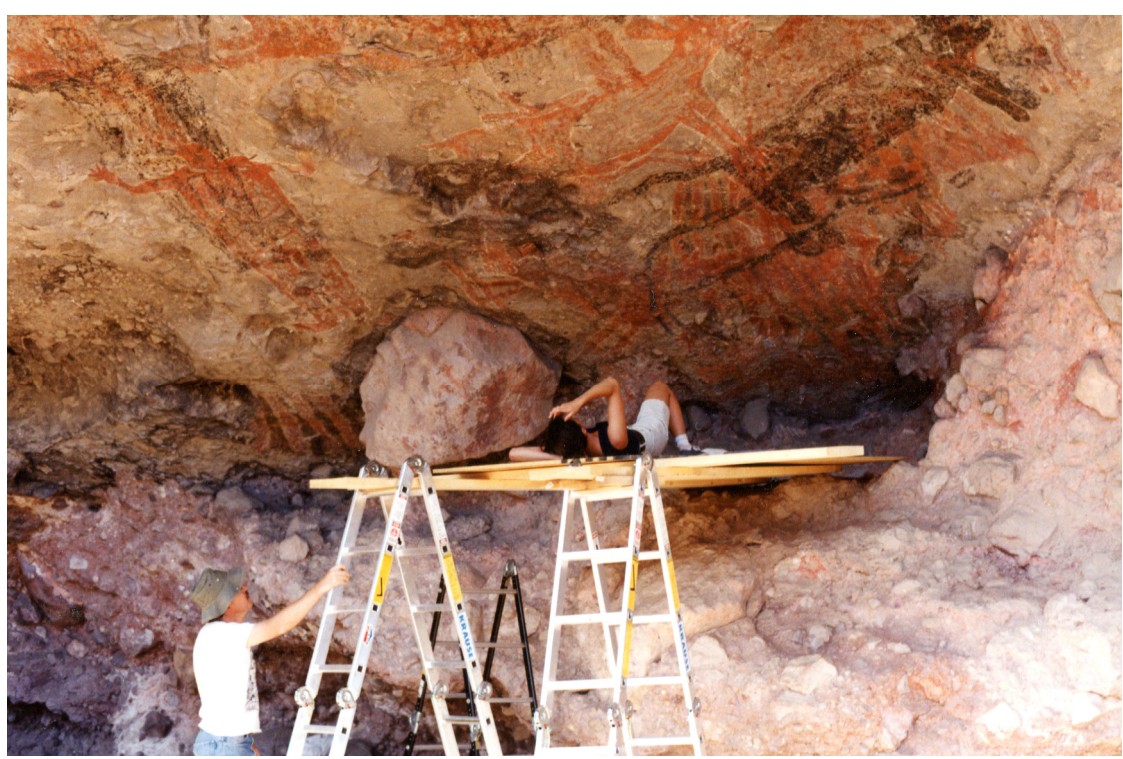

**Figure 8.** Detailed documentation of the Cueva del Ratón site, illustrating complex animal motifs.

The areas showing linear falls in the pH with increasing distance from the back wall reflect the physical dynamics of the site. The most alkaline section is area B, with a mean pH of 5.48 ± 0.62 and this corresponds to a cross section with a depression that might pool water and so promote microbial activity (high in nitrates) and mobilisation of the carbonaceous mineral substrates. Area D, where most of the paintings were found, shows the profile of the protective lip of the shelter and here the mean pH is 4.91 ± 0.33. Area E shows a pronounced overhang, and the mean pH is 4.78 ± 0.20 while the final area F had a slightly more acidic profile at 4.63 ± 0.33. A copy of the plan and cross sections of the site are shown in Figure 9.

The presence of driplines is associated with an increased abundance of the oxalate mineral whewellite since the water mobilises the oxalate metabolites of vegetation which leads to the formation of the black deposits at B1-3, D1-3, and D1-8 are closer to the driplines shown on the site plan. Similar reactions and distributions of the oxalate minerals were found in the arid and calcareous Napier Ranges limestone sites in the Kimberley region of Western Australia. The main difference between the Mexican tuff site and the sites in the Napier Ranges is that the latter received monsoonal rainfall each year. Increased moisture brings about increased microbiological activity and with it oxalates as plant and bacterial metabolites [13,20]. The presence of the silica skins on the site has already been reported [6,10] and has been used to date art on the site, but there was some conjecture as to whether the site was not "wet enough" to create the acidic microenvironment needed to

mobilise silica compounds. However, what must be considered is the quite different chemical reactivity of volcanic tuff formations, compared with the sandstone associated with Australian rock art sites, which have been extensively documented by Alan Watchman [28].

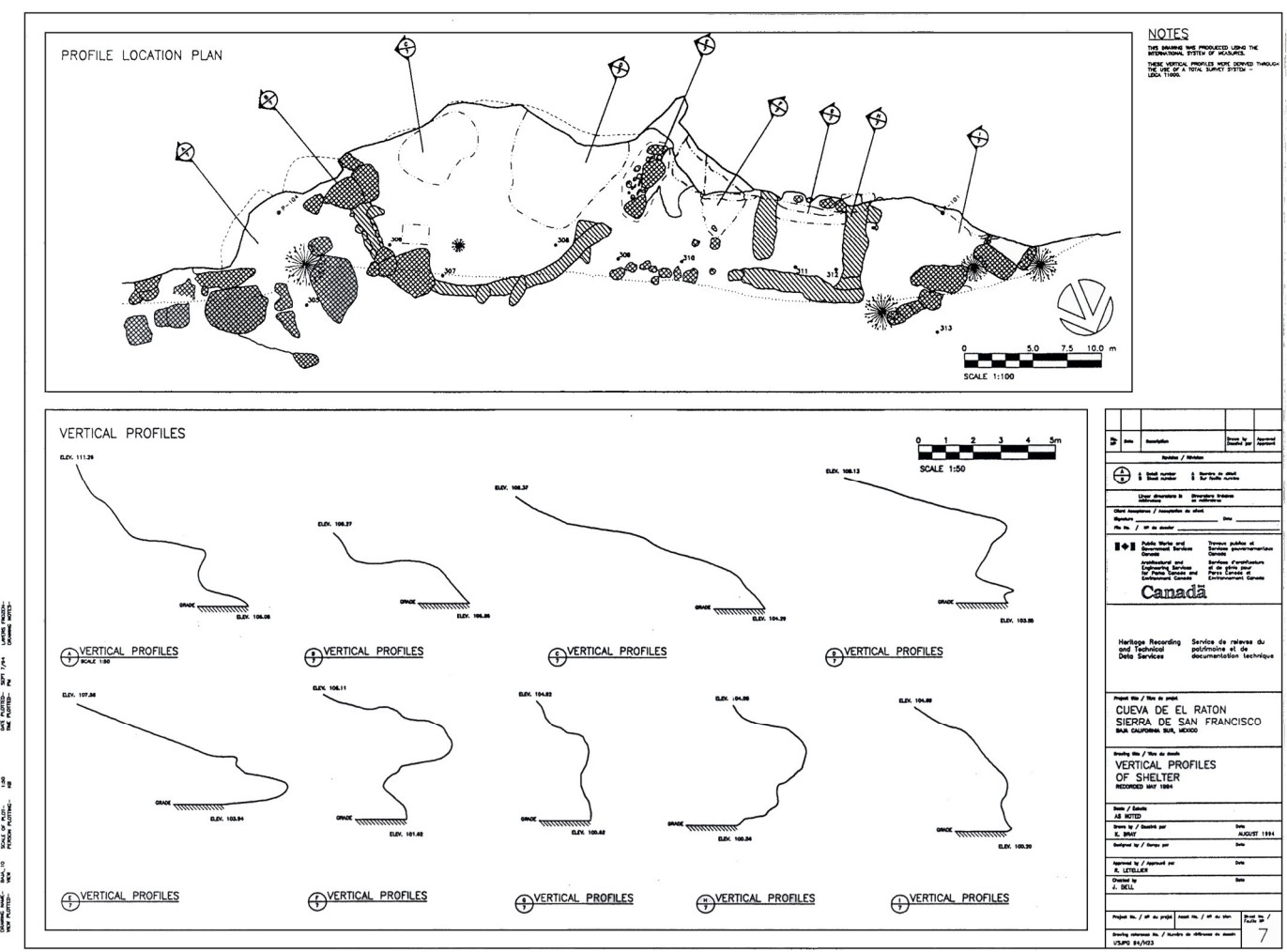

**Figure 9.** Profile location plan and vertical profiles of the site, showing widely different morphology.

The sensitivity of rock art sites and the pigments used in the creative outpourings of the Indigenous communities is well illustrated by the chemical reactions took place during rare rain events (cyclonic depressions) and with the formation of fog at the Walga Rock site in Western Australia and this site has a mean annual rainfall of 232 mm. Such events and the rare rainfall led to mobilisation of the minerals in avian guano deposits high above the artwork which flowed down the rock surface and across the inherently reactive calcitic and kaolinitic pigments. The phosphate minerals in the acidic water brought about mineralogical changes which converted reactive calcite and kaolinite into isomorphous replicates of the original surfaces [29]. The altered pigments are much more stable in terms of their chemical resistance to changes in hydration. Similar fogs form during the winter in the Sierra de San Francisco, and the short distance from the ocean would additionally allow aerosols of sea salts to be deposited on the surface [16,20].

## 7. Conclusions

The measurements undertaken at Cueva del Ratón offer further insight into the alteration and decay mechanisms that are affecting the rock art of the Sierra de San Francisco.

The impact of fogs forming from the contact of the cold sea water on the Pacific Ocean and the hot air coming from the land is important, and the alteration in hydration and the concomitant decay will likely be increased with the changing patterns of rainfall. From the ratio of the chloride to sulphate concentration on the rock surfaces only sites D1-1 and D1-6 had indicative ratios associated with a seaborne source. Our analysis of the distribution of the higher sulphate ion concentrations showed a direct relationship between the sulphate concentration and the surface pH. This data strongly supports the view that for most of the site the sulphate comes from the parent tuff rocks. One of the reasons for survival of the images inside the cave is likely to be associated with the alkaline nature of the water percolating down driplines into the site has functioned as a corrosion and a decay buffer to counterbalance the acidity coming from the microbiological activity associated with the decay of the goat guano. Modelling of the microclimates within such overhangs and on other rock art sites has shown that the range of temperatures and relative humidity can be calculated according to combinations of winds and the arrival and dispersal of moisture fronts [30].

However, the presence of oxalate crusts on the Cueva del Ratón site and the surface concentration of nitrates, with the associated minimum pH values of 4.4, create the appropriate microenvironment for the formation of silica skins and for chemical reactions to take place that result in stable mineral formations [31]. The increased acidity is very likely due to the presence of the layers of goat guano at the mouth of the cave. Biological breakdown of the deposits liberates a range of nitrogenous materials which bacteria, yeasts, moulds, and fungi can use. Metabolites from the living organisms on the site are acidic and this accelerates the alteration of the pigments and both their intra and intramolecular bonding properties. This leads to loss of painted surfaces, either through exfoliation of the substrate or disbondment of the pigments when crystallising minerals under the painted surfaces force off the artworks. Detailed studies on the rock art sites in the California de Sur region along the lines developed in the 2023 paper by Ozan et al. [16] is likely to reveal much needed detail about decay mechanisms that will assist in the conservation management of these culturally significant sites. The formation of reaction products on rock art sites is very much like formation of corrosion products on metals on shipwrecks. The decay reactions are characterised by wet chemical and mineralogical analyses, but we cannot determine the kinetics of their formation. The best way in which the remarkable site at Cueva del Ratón can be assisted is to contract local families to periodically remove the goat guano despite the fencing around the site. There is no evidence of local community engagement associated with the removal of goat guano has taken place but if performed with respect for the archaeological deposits that may lie in the vicinity, such activity would assist in the preservation of the rock art.

Documentation of rock art sites has traditionally been an exercise in applied archaeology, with detailed drawings and photographic recording that also includes multi-spectral imaging, along with geomorphology and mineralogical studies that define the pigments and the substrates. Such information provides the raw data regarding the inherent stability of the paintings and drawings. This work has demonstrated the value of combining such observations with the measurement of surface pH using flat-surface electrodes. The pH or acidity measurements are very sensitive to the microenvironments created by the presence of endemic yeasts, moulds, fungi, and bacteria. Biomineralization processes produce marker minerals such as whewellite, calcium oxalate monohydrate, and this work has provided evidence of the processes that have taken place at times when the local climate was wetter than its current state. A combination of mineralogy and chemistry has opened new possibilities for interpretation of past events. For field studies on a low budget, the use of chloride, nitrate and sulphate test strips can provide additional data that assists

in the interpretation of exfoliation, the formation of mineral accretions and can assist in the identification of the factors controlling the adhesion of mineral pigments to the rock substrates. It is recommended that laboratory-based calibration of the test strips is periodically conducted to connect field values with traditional chemical analyses conducted in laboratory conditions. This information is of great significance in the development of integrated conservation management plans for such sites.

**Author Contributions:** Conceptualization, methodology, validation, writing of draft reports and manuscript I.D.M. and V.M. Project administration and funding acquisition V.M. All authors have read and agreed to the published version of the manuscript.

**Funding:** This research received no external funding.

**Acknowledgments:** The authors gratefully acknowledge the financial assistance of the Instituto Nacional de Antropología e Historia through the Coordinación Nacional de Conservación del Patrimonio Cultural of Mexico for grants that enabled the presentation of a short conservation course in Mexico City and covered the costs of field work.

**Conflicts of Interest:** The authors declare no conflicts of interest.

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
