# Peer review of "Impact of Microbiological Activity and Moisture on the Surface pH of Rock Art Sites: Cueva del Ratón, Baja California Sur, Mexico and Other Sites"

_heritage, doi:10.3390/heritage8090371_

Round 1
Reviewer 1 Report
Comments and Suggestions for Authors
This paper focuses on the important topic of pH and humidity changes at the Cueva del Ratón rock art site, which is part of the Great Mural Tradition of the Baja California Peninsula and is included in the World Heritage List (WHL). It proposes a novel, non-invasive methodology that could help identify the impact of pH and moisture on the development of microbiological communities associated with weathering processes and the formation of accretionary crusts. For this reason, it could make an important contribution to addressing the global issues that affect the preservation of open-air rock art, mainly paintings. This is particularly important given the current scenario of global climate change, which could severely impact the preservation of rock art in the short term. The methodology could offer a rapid diagnosis of salts, humidity and pH. It has been found to correlate significantly with the development of spallation processes and the formation of accretionary crusts.
The procedure suggested in this paper for measuring salts and pH on the surface of rock art panels is remarkable. It is an elegant solution to a complex problem that is often dealt with using expensive equipment, which is not as readily available as test strips and pH meters.
The manuscript is well written. However, it contains several inconsistencies that must be addressed more thoroughly for the study's results and interpretations to be considered robust and reliable.
Firstly, some important references are missing. Notably absent is the book by A. Rubio (2013), which is the most up-to-date and thorough published study of this cave and therefore an unavoidable reference for this paper. The full reference is Rubio i Mora, Albert. 2013. El Yacimiento Arqueológico de El Ratón. Una cueva con pinturas en la Sierra de San Francisco (Baja California Sur, México). II: El mural pintado. Barcelona: Universitat de Barcelona. Seminari d'Estudis i Recerques Prehistòriques.
There are some inconsistencies in the reference [1] to the presence of silica skins in Cueva del Ratón (line 249). This reference does not mention silica skins in this cave or any other. Although it is stated that the art on this site was dated via silica skins, to the best of my knowledge, there is no published bibliography indicating the dating of these accretionary skins at this site. The four samples dated by Fullola et al. in the 1990s provide no information about the substances dated, even though they intended to date the pigment. It is highly likely that these direct radiocarbon dates were affected by the presence of calcium oxalate, which was subsequently dated by Watchman and Gutiérrez in other rock art shelters in the area. It would be necessary to clarify the reference in which was published the results of the silica skin dating, or if it was calcium oxalates.
The structure of the paper is quite inappropriate. It is divided into just three sections: a lengthy introduction; a section that could be considered a combination of results and discussion; and a brief conclusions section. At least a methodology section should be incorporated, detailing the experimental procedure, the equipment used, and the decisions taken to compare some Australian locations with those from Baja California. It is important to provide a rationale for applying some experiments to rocks with different geologies as valid examples for interpreting the results obtained in the Mexican tuff. In this regard, in lines 253–254, criticism is levelled at conjectures about the real possibilities for the formation of silica skins in Baja California shelters, indicating that the geology could explain these formations in less humid conditions. Similarly, it would be appropriate to explain how comparable the Mexican tuff is to granites and limestones from different regions of Australia.
Climatic conditions may be changing in the area due to the northern shift of hurricanes, but this should be supported by climatic data comparing the last few decades to show an increase in precipitation or any relevant data on this topic. The degree of similarity with some Australian regions in South Western Australia and the Burrup Peninsula should also be explained. The climate in Perth is clearly more humid and slightly cooler than in San Francisco de la Sierra; in Murujuga, it is also slightly cooler, but with a similar rainfall pattern. Additional information on this topic is required, as it is a key element in interpreting the results.
Overall, the paper is very positive, but I think these issues should be considered to increase its consistency and impact.
Author Response
There are some inconsistencies in the reference [1] to the presence of silica skins in Cueva del Ratón (line 249). This reference does not mention silica skins in this cave or any other. Although it is stated that the art on this site was dated via silica skins, to the best of my knowledge, there is no published bibliography indicating the dating of these accretionary skins at this site. The four samples dated by Fullola et al. in the 1990s provide no information about the substances dated, even though they intended to date the pigment. It is highly likely that these direct radiocarbon dates were affected by the presence of calcium oxalate, which was subsequently dated by Watchman and Gutiérrez in other rock art shelters in the area. It would be necessary to clarify the reference in which was published the results of the silica skin dating, or if it was calcium oxalates.
The structure of the paper is quite inappropriate. It is divided into just three sections: a lengthy introduction; a section that could be considered a combination of results and discussion; and a brief conclusions section. At least a methodology section should be incorporated, detailing the experimental procedure, the equipment used, and the decisions taken to compare some Australian locations with those from Baja California. It is important to provide a rationale for applying some experiments to rocks with different geologies as valid examples for interpreting the results obtained in the Mexican tuff. In this regard, in lines 253–254, criticism is levelled at conjectures about the real possibilities for the formation of silica skins in Baja California shelters, indicating that the geology could explain these formations in less humid conditions. Similarly, it would be appropriate to explain how comparable the Mexican tuff is to granites and limestones from different regions of Australia.
Climatic conditions may be changing in the area due to the northern shift of hurricanes, but this should be supported by climatic data comparing the last few decades to show an increase in precipitation or any relevant data on this topic. The degree of similarity with some Australian regions in South Western Australia and the Burrup Peninsula should also be explained. The climate in Perth is clearly more humid and slightly cooler than in San Francisco de la Sierra; in Murujuga, it is also slightly cooler, but with a similar rainfall pattern. Additional information on this topic is required, as it is a key element in interpreting the results.
Overall, the paper is very positive, but I think these issues should be considered to increase its
The comments from the reviewer are most helpful and have been addressed in the attached revisions. The key missing reference has been added at number 5. Owing to delays in responding to the comments and in the interest of getting things moving, I will endeavor with my colleague to obtain data pertaining to the increased rainfall events in the area. I am fully aware that the climate in Perth is vastly different to the site in Mexico; the Perth site was used for the purposes of an internal calibration of the test strips. The climate in the Burrup across the now World Heritage Listed Murujuga National Park is arid and hot and the comments on the limestone country in the Kimberley painted sites was given to provide reference material to the friable nature of substrates and the strong impact that weather changes, due to Monsoonal rains, can bring about. Experience in the Burrup within pH and bacterial counts being re-measured within 36 hours of a rare rainfall event showed that very significant changes can occur in a short space of time, following the sudden supply of moisture.
In the meantime I will find the relevant climate data to support the comments about climate changing which can be added in at the second round of review.

Reviewer 2 Report
Comments and Suggestions for Authors
This study offers a valuable, cost-effective model for examining remote sites. Thematically, the article is relevant to the Heritage audience; however, one area that could benefit from improvement is the emphasis on more advanced and precise laboratory methods, as some of the techniques employed are prone to error.
Additionally, incorporating mineral and microbiological analyses could enhance the overall quality of the research.
It is also important to note that issues of generalizability arise when comparing findings to other regions, such as Australia, due to variations in climate and bedrock.
Author Response
The paper has been modified in its structure and the format has been changed to be compliant with the requirements of the reviewers and the journal. Tracked changes are in red and brown, depending on which author made the changes.